# Improvements in Maximum Bite Force with Gum-Chewing Training in Older Adults: A Randomized Controlled Trial

**DOI:** 10.3390/jcm12206534

**Published:** 2023-10-15

**Authors:** Kenta Kashiwazaki, Yuriko Komagamine, Wu Shanglin, Xiangyu Ren, Nanaka Hayashi, Mirai Nakayama, Sahaprom Namano, Manabu Kanazawa, Shunsuke Minakuchi

**Affiliations:** 1Gerodontology and Oral Rehabilitation, Graduate School of Medical and Dental Sciences, Tokyo Medical and Dental University, Tokyo 113-8549, Japan; k.kashiwazaki.gerd@tmd.ac.jp (K.K.); s.wu.gerd@tmd.ac.jp (W.S.); x.ren.gerd@tmd.ac.jp (X.R.); s.namano.gerd@tmd.ac.jp (S.N.); s.minakuchi.gerd@tmd.ac.jp (S.M.); 2Digital Dentistry, Graduate School of Medical and Dental Sciences, Tokyo Medical and Dental University, Tokyo 113-8549, Japan; ma220074@tmd.ac.jp (N.H.); ma220070@tmd.ac.jp (M.N.); m.kanazawa.gerd@tmd.ac.jp (M.K.)

**Keywords:** oral function, older adults, gum-chewing training, randomized controlled trial, maximum bite force

## Abstract

No specific methods have been officially proposed for the prevention and improvement of oral hypofunction. Therefore, in this randomized controlled trial, we aimed to develop a gum-chewing training program and determine its effects in older adults. A total of 218 older adults, aged 65–85 years, were randomly allocated to the intervention or control groups. The intervention group chewed the experimental gum daily, whereas the control group consumed the experimental granular food daily. The outcome assessments measured the maximum bite force, occlusal contact areas, oral dryness, tongue pressure, tongue and lip functions, masticatory function, and gum-chewing time. The measured values for each outcome were compared between groups using the Mann–Whitney U test and within groups pre- and post-intervention using the Wilcoxon signed-rank test. A total of 211 participants completed the study. After 2 months, the intervention group had a significantly higher maximum bite force than the control group (*p* = 0.01), indicating that gum-chewing training improved maximum bite force in older adults. This was determined using one type of bite force measuring device. Therefore, it is suggested that gum-chewing training has a high potential to improve oral hypofunction.

## 1. Introduction

The need for preventive care in the dental field is increasing in developed countries, especially in Japan, which has a rapidly aging population [1]. Preventive care measures in the dental field include the maintenance and improvement of oral function, which are highly associated with a healthy life expectancy [2]. Therefore, diagnostic criteria for the evaluation of oral hypofunction in older adults have been established in Japan [1]. Appropriate diagnosis, management, and motivation can prevent severe oral hypofunction and maintain and restore oral function [3]. Therefore, in the context of Japan’s super-aged society, developing simple and effective intervention methods to maintain and improve oral function is required as a preventive measure for care in the field of dentistry.

Diagnostic methods for oral hypofunction have been continuously adopted since their proposal by the Japanese Society of Geriatric Dentistry in 2016. However, no specific method has been proposed for the prevention and improvement of oral hypofunction. Various interventions have been suggested to improve oral function in older adults [4,5,6,7,8,9,10,11,12,13,14]. Among these methods, chewing gum is a widespread food that is highly palatable, inexpensive, and readily available [15]. Additionally, chewing gum has been shown to reduce stress and improve mood [16]. Therefore, gum-chewing training is considered an easy-to-continue oral training program. In addition, chewing gum is more hygienic and suitable for training because it creates a compact bolus and does not leave any residue in the mouth [17]. Moreover, chewing gum maintains its elasticity for a long period of time, allowing for the continuous loading of the perioral muscles. Therefore, it is expected to be a highly effective training tool.

The authors previously reported the usefulness of chewing training with gum for maintaining and improving oral function in older adults in a preliminary study [18]. In that preliminary study, the effect of gum-chewing training on oral function was evaluated in 130 older adults for one month using a randomized controlled trial. An experimental chewing gum containing granules was used as a training guide for one training session until the texture of the granules was no longer felt by the participant. The results showed that while the maximum bite force, unstimulated saliva flow, tongue and lip functions, and masticatory function were significantly improved in the intervention group in the within-group comparison, only tongue pressure was significantly higher in the between-group comparison, but this was not a large increase. Therefore, to develop a more efficient method for oral function training with a chewing gum containing granules in older adults, it is necessary to further increase the masticatory load. Based on the preliminary study, a new chewing gum for chewing training was developed and the training method was adjusted. First, the hardness and volume of the chewing gum were increased. Second, the training period was changed from one month to two months.

In this study, the authors conducted a randomized controlled trial of chewing training using a newly developed chewing gum. The aim was to examine the effects of chewing training using gum on oral function in older adults. The null hypotheses of this study were that there would be no significant difference in outcomes related to oral function between the intervention and control groups post-intervention and that there would be no significant difference in outcomes related to oral function pre- and post-intervention.

## 2. Materials and Methods

### 2.1. Study Design

This was a double-blind, randomized, controlled trial that adhered to the 2010 Consolidated Standards for Reporting Trials Statement. The protocol in this study was accepted by the Ethics Review Committee of the Tokyo Medical and Dental University (approval number: D2020-063) and registered with the University Hospital Medical Information Network Centre (UMIN-CTR Unique Trial Number: UMIN 000042470).

### 2.2. Participants

The eligibility criteria for the participants were as follows:(1)Individuals aged 65–85 years.(2)Complaints about the inability to masticate stiff foods in comparison to 6 months ago.(3)Independence in daily living activities, including toileting, bathing, walking, eating, and dressing.(4)No previous history of masticatory disorders secondary to muscular disease.(5)No cognitive issues.(6)Did not receive any dental treatment during participation in the trial.(7)At least 20 functional teeth (current number of teeth + number of prosthetic teeth).(8)Able to masticate chewing gum and granular food.(9)No gelatin allergies.

Participants were excluded from this study if:(1)The participant did not meet the eligibility criteria.(2)The participant wanted to retract his/her agreement.(3)Problems that would prevent the study’s continuation occurred during the intervention period.(4)The intake rate of test samples during the intervention period was <80%.(5)The test samples were not consumed for more than five consecutive days.

### 2.3. Intervention

The participants were randomly allocated to the intervention or control groups. Intervention group participants were directed to chew the experimental gum daily for two months. They were directed to chew one piece of the experimental chewing gum twice per set, for a total of three sets daily. The experimental chewing gum was in stick form (36 × 20 × 4.9 mm) and weighed 3.0 g per piece. The experimental chewing gum was designed to be chewed continuously with a constant hardness (7.1 ± 0.5 N), with minimal change in its texture over time. The hardness of the experimental chewing gum was adjusted to be no lower than that of the gum (1.8 N) reported in a study by Nakagawa et al. [5], which significantly increased the biting force. Furthermore, the experimental chewing gum included granules and was designed to enable chewing training as long as all of the granules were crushed. The experimental gum was chewed until the texture of the granules was no longer felt by the participant. The experimental gum was also designed to not stick to dentures. Control group participants were directed to consume the experimental granular food to avoid training effects. They were directed to consume one sachet of the experimental granular foods per set, with three sets per day. In this study, the intervention period was set to two months.

### 2.4. Outcomes

Outcomes were evaluated pre- and post-intervention. Maximum bite force was the primary outcome, while oral dryness, tongue pressure, tongue and lip functions, masticatory function, a food intake questionnaire, and gum-chewing time were the secondary outcomes. The maximum bite force was considered to contribute to the prevention of care needs among several oral functions because Iwasaki et al. [19] reported that a low maximum bite force increased the risk of frailty in older adults in a 5-year longitudinal study. Therefore, the maximum bite force was selected as the primary outcome in this study. Oral function measurements were divided into three sections. In Section 1, the maximum bite force, occlusal contact areas, and unstimulated saliva flow were measured. In Section 2, tongue pressure and tongue and lip functions were measured. In Section 3, masticatory function and gum-chewing time were measured. A 5 min rest period was provided between each section to avoid fatigue.

#### 2.4.1. Maximum Bite Force and Occlusal Contact Areas

The maximum bite force and occlusal contact areas were measured using a pressure-sensitive sheet (Dental Prescale II; GC Co., Tokyo, Japan), as shown in Figure 1. The pressure-sensitive sheet was placed on the occlusal surface and clenched for 3 s in the maximal occlusal position. Then, it was scanned using an analyzer (Bite Force Analyzer; GC Co., Tokyo, Japan) [20,21].

#### 2.4.2. Oral Dryness

Oral dryness was evaluated by measuring the unstimulated saliva flow. Unstimulated saliva was collected in a disposable cup for 2 min and subsequently weighed [22,23].

#### 2.4.3. Tongue Pressure

Tongue pressure was measured using a tongue pressure measurement device (TPM-02; JMS Corporation, Tokyo, Japan), as shown in Figure 2. Participants pressed the balloon of the measuring device with their tongue against the anterior part of their palate with maximum force for 7 s. Measurements were taken three times, and the maximum value was used [24,25,26,27].

#### 2.4.4. Tongue and Lip Functions

Tongue and lip functions were evaluated by calculating the number of times /pa/, /ta/ and /ka/ were pronounced per second. After the participant pronounced each syllable as quickly as possible in 5 s, the automatic measurement device (Kenko-kun Handy; Takei Kiki Kogyo Co., Niigata, Japan), as shown in Figure 3, calculated the number of times each syllable was pronounced per second [28].

#### 2.4.5. Masticatory Function

Masticatory function was objectively evaluated using color-changing chewing gum (Masticatory Performance Evaluating Gum XYLITOL; Lotte Co., Tokyo, Japan), as shown in Figure 4. The gum was chewed 60 times at a rate of once per second. The color-changing chewing gum was then evaluated using the 10-point color scale and a colorimeter (CR-13; KONICA MINOLTA, Tokyo, Japan), and the respective measurements (i.e., color scale value and ΔE value) were obtained [29]. In addition, masticatory function was subjectively evaluated by acquiring the mastication score using a food intake questionnaire [30,31].

#### 2.4.6. Gum-Chewing Time

In order to investigate the actual time spent chewing gum, the gum-chewing time was measured. The time required for each participant to chew the experimental gum was measured twice.

### 2.5. Sample Size

The sample size was calculated using the preliminary study [18]. The mean difference in the maximum bite force pre- and post-training in the intervention group was 98.4 N, and the standard deviation was 241.9 N. Moreover, we set the significance level at 5%, the power at 80%, and the effect size at 0.5. The sample size comprised 96 participants per group. Thus, considering a dropout rate of approximately 10%, this study required 220 participants. The sample size was calculated using R 4.0.0 (The R Foundation for Statistical Computing, Vienna, Austria).

### 2.6. Randomization and Blinding

The study participants were randomly assigned to individuals who were not evaluators using a stratified block method with sex and age as stratification factors. Moreover, the participants were not informed about what would be distributed as test samples, and the test samples were placed in a silver bag by individuals who were not evaluators. Therefore, the evaluators in the study did not know whether the experimental gum or granular foods were contained in the bag. Thus, this was a double-blind study.

### 2.7. Statistical Analysis

For between-group comparisons, the chi-square test was used for sex and number of denture wearers, with the *t*-test for height and weight and the Mann-Whitney U test for the other data. For within-group comparisons, the Wilcoxon sign-rank test was used. SPSS software (version 23.0; IBM Corp., Armonk, NY, USA) was used for all statistical analyses. Statistical significance was set at a *p*-value < 0.05.

## 3. Results

### 3.1. Participants

Figure 5 shows a flow diagram of the study participants. At the start of the intervention, 218 participants were enrolled, and 7 of them were excluded for meeting the exclusion criteria. A total of 211 participants completed the study. The baseline characteristics are presented in Table 1. There were no significant differences in the baseline characteristics of the 107 intervention group participants and the 104 control group participants.

### 3.2. Measurement of Oral Function

The results for each outcome are presented in Table 2 for between-group comparisons and in Table 3 for within-group comparisons pre- and post-intervention.

The maximum bite force was significantly higher in the intervention group compared to the control group post-intervention (*p* = 0.01). The within-group comparisons showed a significant increase in the intervention group (*p* < 0.01).

The occlusal contact areas were not significantly different between the intervention and control groups, both pre- and post-intervention. The within-group comparisons showed a significant increase in the intervention group (*p* < 0.01).

Unstimulated salivary flow was not significantly different between the intervention and control groups, either pre- or post-intervention. The within-group comparisons showed no significant increase in either group.

Tongue pressure was not significantly different between the intervention and control groups, both pre- and post-intervention. The within-group comparisons showed significant increases in the intervention and control groups (intervention group, *p* < 0.01, control group, *p* = 0.02).

Tongue and lip motor functions were not significantly different between the intervention and control groups, both pre- and post-intervention. The within-group comparisons revealed no significant increase in either group.

Masticatory function (color scale and ΔE value) did not differ significantly between the intervention and control groups, both pre- and post-intervention. The within-group comparisons showed significant increases in the intervention and control groups (color scale: *p* < 0.01, both groups; ΔE: *p* < 0.01, both groups). Subjective masticatory function (mastication score) was not significantly different between the intervention and control groups, both pre- and post-intervention. The within-group comparisons revealed significant increases in the intervention and control groups (p < 0.01).

The gum-chewing time was not significantly different between the intervention and control groups, both pre- and post-intervention. However, for the within-group comparison, the gum-chewing time was significantly reduced in the intervention group (*p* = 0.03).

## 4. Discussion

In this study, the null hypothesis was rejected for maximum bite force in the between-group comparison. Additionally, for the within-group comparisons of the intervention groups, the null hypotheses were rejected for maximum bite force, occlusal contact areas, tongue pressure, masticatory function (color scale, ΔE value, and mastication score), and gum-chewing time.

Maximum bite force, which was the primary outcome of this study, was significantly higher in the intervention group than in the control group. Although Kim et al. [14] previously evaluated the effect of gum-chewing training on oral function in older adults in a randomized controlled trial, this study is the first to show a significantly higher maximum bite force in the intervention group post-intervention compared to pre-intervention. The results of this study also differ from those of the previously conducted preliminary study [18]. Here, several improvements were made to the gum training in this study. First, the gum hardness was significantly increased compared to that in the preliminary study (preliminary study: 2.7 ± 0.2 N; this study: 7.1 ± 0.5 N), potentially generating a sufficient training load to improve the maximum bite force. Second, the intervention period was extended from one to two months, following a similar approach to that taken by Kim et al. [14]. Our results suggest that the extension of the intervention period from one to two months was an appropriate approach to ensure a sufficient training time. Previous studies have suggested that maximum bite force is highly related to the stiffness of the masseter muscle [32]. Therefore, the significantly higher maximum bite force in the intervention group in the between-group comparison after the intervention in this study could be due to an increase in masseter muscle stiffness (functional changes due to increased masseter muscle contractile capacity) after chewing training. In addition, in this study, Dental Prescale II was used for the maximum bite force measurement. The reliability of Dental Prescale II has been established in previous studies [33,34,35]. Wang et al. [33] compared the reliability of the occlusion assessment using three different occlusion measurement tools articulating paper, Dental Prescale II, and a virtual occlusal construction method, with that using a silicone bite registration material and reported that all three, including Dental Prescale II, showed a good reliability in clinical applications. In addition, Dental Prescale II showed significant correlations with the other bite-force-measuring devices, Dental Prescale (the previous version of the Dental Prescale II) and T-Scan, with a high reliability [34,35]. Therefore, Dental Prescale II is considered to be a tool that can be used to quickly and simultaneously evaluate both occlusal bite force and occlusal contact area. 

In this study, only two intervention group participants dropped out after the start of the intervention, and no adverse events were associated with the gums used. Therefore, a training method that does not require the self-measuring of chewing time or the amount of chewing, as in this study, is considered a simple, easy-to-continue, and safe training method for older adults, which might lead to fewer dropouts. Furthermore, unlike other food samples, chewing gum does not fall apart, forms a compact bolus after chewing, and can be completely removed from the oral cavity, making it an excellent sample for oral hygiene purposes. In addition, 31% of the intervention group participants in this study were denture wearers. The gum in this study was designed to allow for chewing of the gum without denture detachment or overturning, and few participants dropped out, suggesting the chewing gum is suitable as a training method for denture wearers.

In the within-group comparisons, a significant increase in tongue pressure was observed in the intervention groups. Previous studies have reported that chewing training with chewing gum and gummy jellies increases tongue pressure [10,12], which is consistent with the results of this study. In addition, a significant increase in masticatory function was observed in the intervention group. Various factors related to masticatory function when using color-changing gums have been reported [36,37,38,39,40,41,42,43]. Among these relevant factors, in this study, the maximum bite force in the intervention group increased in the between-group comparison after the intervention, suggesting that the maximum bite force may have played a significant role in the improvement of masticatory function in the within-group comparison of the intervention group. Furthermore, the occlusal contact areas only increased significantly in the intervention group. However, the occlusal contact areas of participants who did not receive dental treatment were unlikely to increase during the short two-month intervention period. Dental Prescale II, which was used to measure the maximum bite force and occlusal contact areas in this study, was designed to measure the area of red ink in the pressure-sensitive sheet that is ejected when pressure from a bite force is applied to the capsule contained inside the sheet. Therefore, the occlusal contact areas may also have increased in this study because more capsules burst with the increase in the maximum bite force.

The within-group comparisons revealed significant increases in tongue pressure and the objective and subjective masticatory functions in the control group. However, it is unlikely that tongue pressure and masticatory function would be improved by the continuous intake of granular foods. The significant increase in these outcomes may have been due to a placebo effect. A placebo effect is an observed improvement due to the patient’s belief that they are receiving a real drug, even if a fake drug is prescribed [44,45]. In this study, it is possible that the control group participants assumed that the experimental granular foods were effective in maintaining and improving oral function.

This study had several limitations. First, although one of the eligibility criteria for this study was to recruit participants who complained of difficulty masticating stiff foods compared to 6 months ago, the median outcome for each oral function before the intervention was not significantly lower than the reference value. Therefore, it is necessary to demonstrate the effects of gum-chewing training on oral function in individuals who have been diagnosed with oral hypofunction. Second, whether or not the participants completed the two-month training was determined using their diaries in the network system. However, because we thus depended on the participants themselves to carry out the training, it was difficult to determine whether the training was followed and continued. Third, although the reliability of the Dental Prescale II was confirmed in previous studies [33,34,35], the measurements should be performed using two or more other types of devices for accuracy. Furthermore, the maximum bite force should be measured together with an electromyogram to confirm that each participant clenched at the maximum bite force [46]. Because this study was a large survey, it was not possible to spend much time on each measurement, and only one type of bite force measuring device, the Dental Prescale II, was used.

In the future, the gum-chewing training procedure performed in this study holds potential to maintain and improve oral function in older adults through the widespread use of this training method. Maximum bite force might be one of the diagnostic parameters for oral hypofunction, and it has been reported that a low maximum bite force increases the risk of frailty in older adults [18]. The significant differences in maximum bite force found in this study suggest its high possibility of contributing to the improvement of oral hypofunction and prevention of care in a super-aged society.

## 5. Conclusions

Within the limitations of this study, the maximum bite force using one type of bite force measuring device in the intervention group was significantly higher compared to the control group after two months of gum-chewing training based on the concept of chewing until the texture of the granules in the gum could no longer be sensed.

## Figures and Tables

**Figure 1 jcm-12-06534-f001:**
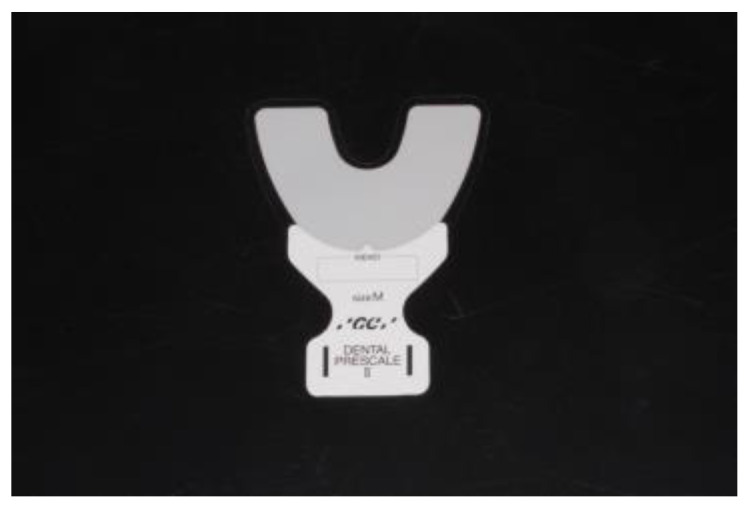
Maximum bite force and occlusal contact area measurement device (Dental Prescale II; GC Co., Tokyo, Japan).

**Figure 2 jcm-12-06534-f002:**
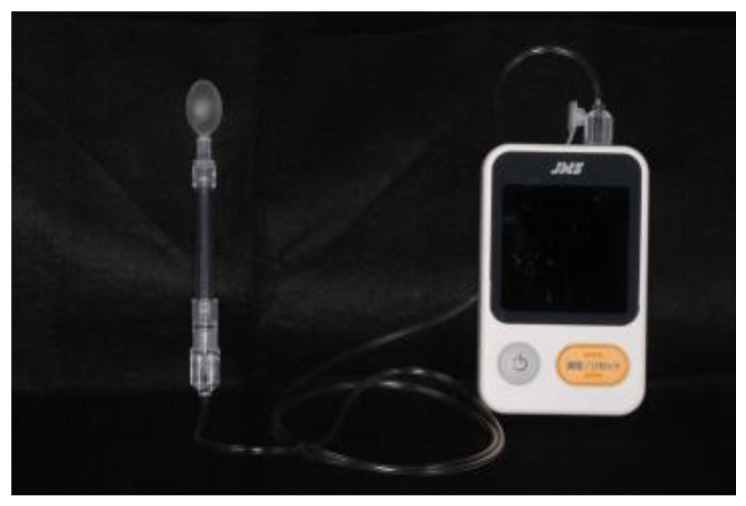
Tongue pressure measurement device (TPM-02; JMS Corporation, Tokyo, Japan).

**Figure 3 jcm-12-06534-f003:**
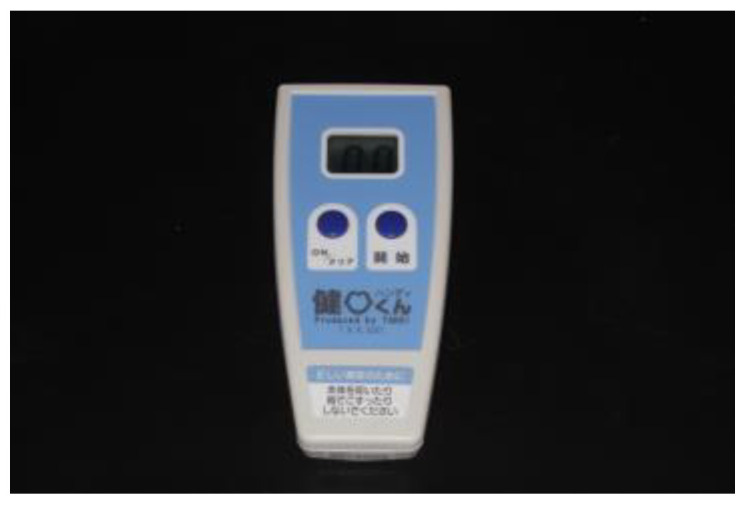
Tongue and lip function measurement device (Kenko-kun Handy; Takei Kiki Kogyo Co., Niigata, Japan).

**Figure 4 jcm-12-06534-f004:**
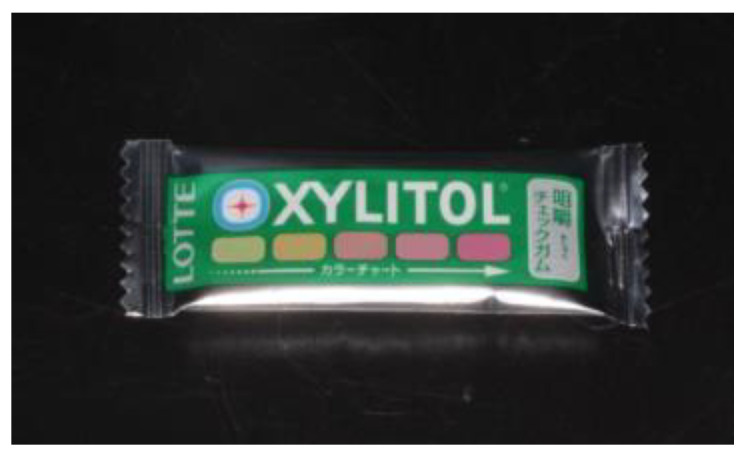
Color-changing chewing gum (Masticatory Performance Evaluating Gum XYLITOL; Lotte Co., Tokyo, Japan).

**Figure 5 jcm-12-06534-f005:**
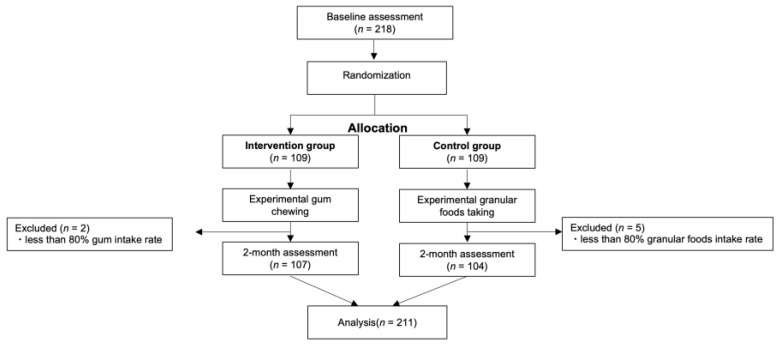
Consolidated Standards of Reporting Trials (CONSORT) flow chart of study participants.

**Table 1 jcm-12-06534-t001:** Baseline characteristics of study participants.

	Intervention (*n* = 107)	Control (*n* = 104)	*p*-Value
Age, years ^a^	70.0 (7.0)	69.0 (6.0)	0.97 ^b^
Sex (%)			0.83 ^e^
Female	53 (49%)	50 (48%)	
Male	54 (51%)	54 (52%)	
Body height, cm ^c^	161.3 (8.4)	162.6 (8.4)	0.25 ^d^
Body weight, kg ^c^	59.5 (11.4)	60.4 (11.7)	0.58 ^d^
Number of teeth ^a^	26.0 (4.0)	26.0 (3.0)	0.57 ^b^
Number of functional teeth ^a^	28.0 (1.0)	28.0 (1.0)	0.69 ^b^
Number of denture wearers	33 (31%)	29 (28%)	0.64 ^e^

^a^ Data are presented as medians (interquartile range). ^b^ Based on the Mann–Whitney U test. ^c^ Data are presented as the mean (standard deviation). ^d^ Based on the *t*-test. ^e^ Based on the chi-square test.

**Table 2 jcm-12-06534-t002:** Between-group differences in the median (interquartile range) baseline and 2-month post-intervention results of oral functions.

	Baseline			2M		
	Intervention (*n* = 107)	Control (*n* = 104)	*p*-Value	Intervention (*n* = 107)	Control (*n* = 104)	*p*-Value
Maximum bite force (N)	457.8 (442.1)	512.1 (447.8)	0.25	663.8 (509.1)	507.0 (411.3)	0.013 *
Occlusal contact areas (mm^2^)	11.8 (13.1)	12.8 (13.2)	0.28	16.2 (14.4)	15.0 (10.9)	0.08
Unstimulated saliva flow (g)	1.8 (1.5)	1.7 (1.8)	0.53	1.7 (1.4)	1.8 (1.6)	0.06
Tongue pressure (kPa)	35.5 (11.2)	33.7 (9.6)	0.12	37.6 (12.9)	36.2 (8.0)	0.21
Tongue and lip functions						
/pa/ (times)	6.2 (1.0)	6.0 (1.0)	0.94	6.2 (1.0)	6.4 (1.0)	0.33
/ta/ (times)	6.2 (1.0)	6.2 (1.0)	0.92	6.2 (1.0)	6.2 (1.0)	0.69
/ka/ (times)	5.8 (1.0)	5.6 (1.0)	0.42	5.8 (1.0)	5.8 (1.0)	0.96
Color-changing chewing gum						
Color scale	9.0 (2.0)	9.0 (2.0)	0.33	10.0 (1.0)	10.0 (1.0)	0.63
ΔE	48.2 (5.8)	47.0 (6.5)	0.27	49.2 (6.6)	50.4 (6.1)	0.24
Mastication score	90.3 (14.0)	91.5 (15.0)	0.40	93.5 (14.0)	94.8 (13.0)	0.28
Masticatory time (mins)	4.7 (1.9)	4.5 (1.8)	0.30	4.4 (2.2)	4.5 (2.6)	0.30

2M, 2-month assessment. Data are presented as median (interquartile range). * Significant difference (*p* < 0.05).

**Table 3 jcm-12-06534-t003:** Within-group differences in the median (interquartile range) baseline and 2-month post-intervention results of oral functions.

	Intervention (*n* = 107)			Control (*n* = 104)		
	Baseline	2M	*p*-Value	Baseline	2M	*p*-Value
Maximum bite force (N)	457.8 (442.1)	663.8 (509.1)	<0.001 *	512.1 (447.8)	507.0 (411.3)	0.97
Occlusal contact areas (mm^2^)	11.8 (13.1)	16.2 (14.4)	<0.001 *	12.8 (13.2)	15.0 (10.9)	0.10
Unstimulated saliva flow (g)	1.8 (1.5)	1.7 (1.4)	0.85	1.7 (1.8)	1.8 (1.6)	0.79
Tongue pressure (kPa)	35.5 (11.2)	37.6 (12.9)	<0.001 *	33.7 (9.6)	36.2 (8.0)	0.021 *
Tongue and lip functions						
/pa/ (times)	6.2 (1.0)	6.2 (1.0)	0.30	6.0 (1.0)	6.4 (1.0)	0.50
/ta/ (times)	6.2 (1.0)	6.2 (1.0)	0.88	6.2 (1.0)	6.2 (1.0)	0.92
/ka/ (times)	5.8 (1.0)	5.8 (1.0)	0.22	5.6 (1.0)	5.8 (1.0)	0.70
Color-changing chewing gum						
Color scale	9.0 (1.0)	10.0 (1.0)	<0.001 *	9.0 (2.0)	10.0 (1.0)	<0.001 *
ΔE	48.2 (5.8)	49.2 (6.6)	<0.001 *	47.0 (6.5)	50.4 (6.1)	<0.001 *
Mastication score	90.3 (14.0)	93.5 (14.0)	<0.001 *	91.5 (15.0)	94.8 (13.0)	<0.001 *
Masticatory time (mins)	4.7 (1.9)	4.4 (2.2)	0.031 *	4.5 (1.8)	4.5 (2.6)	0.82

2M, 2-month assessment. Data are presented as medians (interquartile ranges). * Significant difference (*p* < 0.05).

## Data Availability

The data sets generated and analyzed during the current study are not publicly available due to the data confidentiality requirements of the ethics committee but are available from the corresponding author on reasonable request and approval from the ethics committee.

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
