# Peer review of "Improvements in Maximum Bite Force with Gum-Chewing Training in Older Adults: A Randomized Controlled Trial"

_jcm, 2023, doi:10.3390/jcm12206534_

Round 1

Reviewer 1 Report (Previous Reviewer 1)

Dear authors,

Thank you for the revision sent.

Therefore, I have one more question. "The maximum bite force and occlusal contact area were measured using a pressure-sensitive sheet (Dental Prescale II; GC Co., Tokyo, Japan) and an analyzer (Bite Force Analyzer; GC Co., Tokyo, Japan)."

What is the reliability of the bite registered using a sheet? I suggest including another test to improve the results using another technique.

Author Response

Reply to reviewer #1

As indicated in the responses that follow, we have taken considered all comments and suggestions in the revised version of our paper.

Thank you for the revision sent.

Therefore, I have one more question. "The maximum bite force and occlusal contact area were measured using a pressure-sensitive sheet (Dental Prescale II; GC Co., Tokyo, Japan) and an analyzer (Bite Force Analyzer; GC Co., Tokyo, Japan)."

What is the reliability of the bite registered using a sheet? I suggest including another test to improve the results using another technique.

Answer:

Thank you for your important suggestion. As you point out, in this study, the Dental Prescale II was used for the maximum bite force measurement. While the thickness of an articulating paper is 30 to 80 μm, the one of the Dental Prescale II is 150 μm, which might be less reproducible in terms of occlusal contact area and number of occlusal contact points in occlusion. However, in a previous study, the reliability and validity of evaluation for occlusal contact area and occlusal contact numbers using three different occlusion measurement tools, an articulating paper, Dental Prescale II and the virtual occlusal construction method, were compared with those using silicon impression material. The results reported that the three different occlusion measurement tools showed good reliability and validity for clinical application. Therefore, Dental Prescale II is considered to be a tool that can quickly and simultaneously evaluate both occlusal bite force and occlusal contact area in occlusion. This information has been added in the second paragraph of the discussion.

Reviewer 2 Report (New Reviewer)

Thank you for your manuscript. It is a very important topic. 

I would like to ask you to give more informations about the kind of chewing gum you used. I would suggest to add also an image if posssible. 

 i would add immages about these devices that you cite. 

- device (TPM-02; 141 JMS Corporation, Tokyo, Japan).

-device (Kenko-kun Handy; Takei Kiki Kogyo ...)

I would suggest to wirte something about the anamnesis of these patients, if they have other disease that can condition their chewing capability. Also i would see if they suffer of bruxism or not.

Thank you

Author Response

Reply to reviewer #2

As indicated in the responses that follow, we have taken considered all comments and suggestions in the revised version of our paper.

Answer:

Thank you for your manuscript. It is a very important topic. 

I would like to ask you to give more informations about the kind of chewing gum you used. I would suggest to add also an image if posssible. 

 i would add immages about these devices that you cite. 

- device (TPM-02; 141 JMS Corporation, Tokyo, Japan).

-device (Kenko-kun Handy; Takei Kiki Kogyo ...)

I would suggest to wirte something about the anamnesis of these patients, if they have other disease that can condition their chewing capability. Also i would see if they suffer of bruxism or not.

Thank you

Thank you for your important suggestion. Any more details about the experimental chewing gum can’t be described. However, I have added the figures of measurement devices used in this study, including the devices you pointed out.

In addition, we have excluded the participants with masticatory disorder and muscular disease that might influence masticatory function, although we did not examine the presence or absence of bruxism. This information has been mentioned in the 2.2. Participants section of Materials and Methods. Therefore, it is unlikely that the presence or absence of bruxism directly influenced the masticatory function in this study.

Round 2

Reviewer 1 Report (Previous Reviewer 1)

Dear authors,

Please add another method to show the reliability of the data obtained in your study. The response to my question showed that the material used was not sufficient.

Thank you.

Author Response

Reply to reviewer #1

Dear authors,

Please add another method to show the reliability of the data obtained in your study. The response to my question showed that the material used was not sufficient.

Thank you.

Answer:

Thank you for your important suggestion. The Dental Prescale II used in this study established its reliability in previous studies, and significant correlations with the other bite force measuring device with high reliability were reported. However, the measurement should be performed using two or more types of devices for accuracy. Furthermore, maximum bite force should be measured together with an electromyogram to confirm that each participant clenched at maximum bite force. Because this study was a large survey, it was not possible to spend much time on each measurement, only one type of bite force measuring device, the Dental Prescale II, was selected. This information has been added in the second and sixth paragraphs of the Discussion in the manuscript.

Reviewer 2 Report (New Reviewer)

Thank you for the corrections and your attempt to improve the manuscript. 

I would suggest to increase the references and the description of “ dental prescale II”.

thank you

English seems fine

Author Response

Reply to reviewer #2

Thank you for the corrections and your attempt to improve the manuscript. 

I would suggest to increase the references and the description of “dental prescale II”.

 thank you

Answer:

Thank you for your important suggestion. We have added references for Dental Prescale II (the reference numbers are 33 to 35). In addition, we have added the description of Dental Prescale II in the second and sixth paragraphs of the Discussion in the manuscript.

This manuscript is a resubmission of an earlier submission. The following is a list of the peer review reports and author responses from that submission.

Round 1

Reviewer 1 Report

It was evaluated the article titled "Improvement of maximum bite force by gum-chewing training in older adults: A randomized controlled trial”. The goal was "to develop a gum-chewing training program and clarify its effects in older adults”.

The study is interesting, and the topic is important. Otherwise, the pilot study published was better than this.
It is well-written.

- “The sample size was calculated from the preliminary study [18].”
When I checked the study, they enrolled 130 patients; the final analysis had 117 patients.

For this study, the aim was 192 participants. Therefore, only 117 final analyses were made. It is extremely below the goal. The pilot study was better. Why do the authors want to publish these incomplete results?

Hence, it is not possible to draw any conclusion.

This article is weaker than the pilot.